# Realizing a deep reinforcement learning agent for real-time quantum feedback

Kevin Reuer[1,2] ✉, Jonas Landgraf[3,4], Thomas Fösel[3,4], James O'Sullivan[1,2], Liberto Beltrán[1,2], Abdulkadir Akin[1,2], Graham J. Norris[1,2], Ants Remm[1,2], Michael Kerschbaum[1,2], Jean-Claude Besse[1,2], Florian Marquardt[3,4], Andreas Wallraff[1,2] & Christopher Eichler[1,5] ✉

Realizing the full potential of quantum technologies requires precise real-time control on time scales much shorter than the coherence time. Model-free reinforcement learning promises to discover efficient feedback strategies from scratch without relying on a description of the quantum system. However, developing and training a reinforcement learning agent able to operate in real-time using feedback has been an open challenge. Here, we have implemented such an agent for a single qubit as a sub-microsecond-latency neural network on a field-programmable gate array (FPGA). We demonstrate its use to efficiently initialize a superconducting qubit and train the agent based solely on measurements. Our work is a first step towards adoption of reinforcement learning for the control of quantum devices and more generally any physical device requiring low-latency feedback.

Executing algorithms on future quantum information processing devices will rely on the ability to continuously monitor the device's state via quantum measurements and to act back on it, on timescales much shorter than the coherence time, conditioned on prior observations[1,2]. Such real-time feedback control of quantum systems, which offers applications e.g. in qubit initialization[3–6], gate teleportation[7,8] and quantum error correction[9–11], typically relies on an accurate model of the underlying system dynamics. With the increasing number of constituent elements in quantum processors such accurate models are in many cases not available. In other cases, obtaining an accurate model will require significant theoretical and experimental effort. Model-free reinforcement learning[12] promises to overcome such limitations by learning feedback-control strategies without prior knowledge of the quantum system.

Reinforcement learning has had success in tasks ranging from board games[13] to robotics[14]. Reinforcement learning has only very recently been started to be applied to complex physical systems, with training performed either on simulations[15–21] or directly in experiments[22–29], for example in laser[22,25,29], particle[23,24], soft-matter[26]

and quantum physics[27,28]. Specifically in the quantum domain, during the past few years, a number of theoretical works have pointed out the great promises of reinforcement learning for tasks covering state preparation[30–34], gate design[35], error correction[36–38], and circuit optimization/compilation[39,40], making it an important part of the machine learning toolbox for quantum technologies[41–43]. In first applications to quantum systems, reinforcement learning was experimentally deployed, but training was mostly performed based on simulations, specifically to optimize pulse sequences for the quantum control of atoms and spins[17,18,21]. Beyond that, there are two pioneering works demonstrating the training directly on experiments[27,28] which was used to optimize pulses for quantum gates[27] and to accelerate the tune-up of quantum dot devices[28]. However, none of these experiments[17,18,21,27,28] featured real-time quantum feedback. Real-time quantum feedback is crucial for applications like fault-tolerant quantum computing[44]. Realizing it using deep reinforcement learning in an experiment has remained an important open challenge. Very recently, a step into this direction was made in ref. 45, which demonstrates the use of reinforcement learning for quantum error correction. In

[1]Department of Physics, ETH Zurich, CH-8093 Zurich, Switzerland. [2]Quantum Center, ETH Zurich, CH-8093 Zurich, Switzerland. [3]Max Planck Institute for the Science of Light, Staudtstraße 2, 91058 Erlangen, Germany. [4]Physics Department, University of Erlangen-Nuremberg, Staudtstraße 5, 91058 Erlangen, Germany. [5]Present address: Physics Department, University of Erlangen-Nuremberg, Staudtstraße 5, 91058 Erlangen, Germany. ✉e-mail: kevin.reuer@phys.ethz.ch; christopher.eichler@fau.de

contrast to what we present in this paper, these experiments[45] relied on searching for the optimal parameters of a controller with fixed structure.

Here, we realize a reinforcement learning agent that interacts with a quantum system on a sub-microsecond timescale. This rapid response time enables the use of the agent for real-time quantum feedback control. We implement the agent using a low-latency neural network architecture, which processes data concurrently with its acquisition, on a field-programmable gate array (FPGA). As a proof of concept, we train the agent using model-free reinforcement learning to initialize a superconducting qubit into its ground state without relying on a prior model of the quantum system. The training is performed directly on the experiment, i.e., by acquiring experimental data with updated neural network parameters in every training step. In repeated cycles, the trained agent acquires measurement data, processes it and applies pre-calibrated pulses to the qubit conditioned on the measurement outcome until the agent terminates the initialization process. We study the performance of the agent during training and demonstrate convergence in less than three minutes wall clock time, after training on less than 30,000 episodes. Furthermore, we explore the strategies of the agent in more complex scenarios, i.e. when performing weak measurements or when resetting a qutrit.

## Results
### Reinforcement learning for a qubit
In model-free reinforcement learning, an agent interacts with the world around it, the so-called reinforcement learning environment (Fig. 1). In repeated cycles, the agent receives observations **s** from the environment and selects actions $a$ according to its policy $\pi$ and the respective observation **s**. In the important class of policy-gradient methods[12], this policy is realized as a conditional probability distribution $\pi_\theta(a|s)$, which can be modeled as a neural network with parameters $\theta$. To each sequence of observation-action pairs, called an episode, one assigns a cumulative reward $R$. The goal of reinforcement learning is to maximize the reward $\bar{R}$ averaged over multiple episodes, by updating the parameters $\theta$ e.g. via gradient ascent $\Delta\theta \sim \nabla_\theta \bar{R}$[12]. Such a policy-gradient procedure is able to discover an optimal policy even without access to an explicit model of the dynamics of the reinforcement learning environment.

In the present work, we use reinforcement learning to learn strategies for real-time control of quantum systems. Here, observations are obtained via quantum measurements, actions are realized as unitary gate operations, and the reward is measured in terms of the speed

and fidelity of initializing the quantum system into a target state, see schematic in Fig. 1. In our experiment, the quantum system is realized as a transmon qubit with ground $|g\rangle$, excited $|e\rangle$, and second excited state $|f\rangle$ dispersively coupled to a superconducting resonator[46] (see Supplementary Note 1 for details). We probe the qubit with a microwave field, which scatters off the resonator and is amplified and digitized to result in an observation vector **s** = (**I**, **Q**), where **I** and **Q** are time traces of the two quadrature components of the digitized signal[47–49] (see Supplementary Note 2 for details and Supplementary Note 3 for averaged time traces). Depending on **s** the agent selects, according to its policy $\pi$, one of several discrete actions in real time. In the simplest case, it either idles until the next measurement cycle, it performs a bit-flip as a unitary swap between $|g\rangle$ and $|e\rangle$ or it terminates the initialization process.

To train the agent, we transfer batches of episodes to a personal computer (PC) serving as a reinforcement learning trainer. The reinforcement learning trainer computes the associated reward for each episode and updates the agent's policy accordingly (see Supplementary Note 4 for details), before returning the updated network parameters $\theta$ to the FPGA.

### Implementation of the real-time agent
We implement this scheme in an experimental setup, in which the agent, for each episode, can perform multiple measurement cycles $j$, in each of which it receives a qubit-state-dependent observation **s**$^j$ and selects an action $a^j$, until it terminates the episode, see Fig. 2a. If the agent selects the bit-flip action, a $\pi$-pulse is applied to the qubit after a total latency of $\tau_{EL,tot}$ = 451 ns, dominated by analog-to-digital and digital-to-analog converter delays. The agent's neural network contributes only $\tau_{NN}$ = 48 ns to the total latency as it is evaluated mostly during qubit readout and signal propagation (see Supplementary Note 2 for detailed discussion of the latency). To provide the agent with a memory about past cycles we feed downsampled

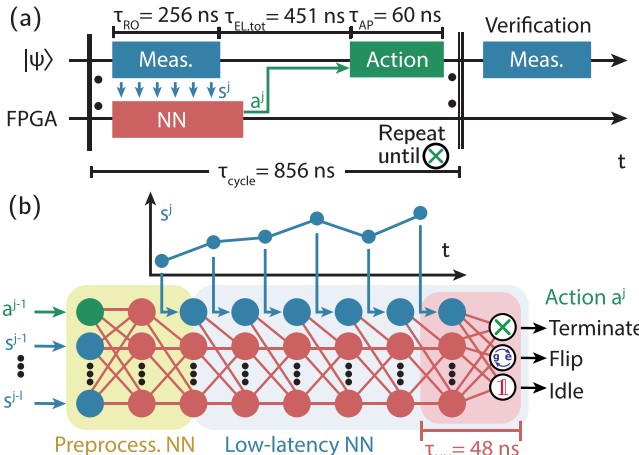

**Fig. 2 | Schematic of neural-network-based real-time feedback control. a** Timing diagram of an experimentally realized reinforcement learning episode. In each cycle $j$, the observation **s**$^j$ resulting from a measurement (Meas., blue) is continuously fed into a neural network (NN, red) which determines the next action $a^j$ (green). The process is terminated after a number of cycles determined by the agent. Then, a verification measurement is performed. **b** Schematic of the neural network implemented on an FPGA. The neural network consists of fully connected (red lines) layers of feed-forward neurons (red dots) and input neurons (blue dots for observations, green dots for actions). The first layers form the preprocessing network (yellow background). During the evaluation of the low-latency network (blue background), new data points from the signal trace **s**$^j$ are fed into the network as they become available. The network outputs the action probabilities for the three actions. Only the execution of the last layer (red background) contributes to the overall latency.

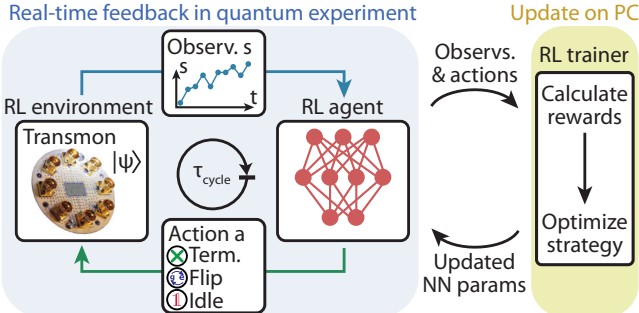

**Fig. 1 | Concept of the experiment.** A reinforcement learning (RL) agent, realized as a neural network (NN, red) on a field-programmable gate array (FPGA), receives observations **s** (Observ., blue trace) from a quantum system, which constitutes the reinforcement learning environment. Here, the quantum system is realized as a transmon qubit coupled to a readout resonator fabricated on a chip (see photograph). The agent processes observations on sub-microsecond timescales to decide in real time on the next action $a$ applied to the quantum system. The update of the agent's parameters is performed by processing experimentally obtained batches of observations and actions on a PC.

observations ($\mathbf{s}^{j-1}, \ldots, \mathbf{s}^{j-l}$) and actions ($a^{j-1}, \ldots, a^{j-l}$) from up to $l = 2$ previous cycles into the neural network. To characterize the performance of the agent, we perform a verification measurement $\mathbf{s}^{\text{ver}}$ after termination.

Any neural-network agent used for real-time system control greatly benefits from short latencies in the signal processing. For our FPGA implementation we therefore introduce a network architecture, in which new measurement data is processed as soon as it becomes available, thereby keeping latencies at a minimum. More specifically, we sequentially feed elements $I_k^j, Q_k^j$ of the digitized time trace $\mathbf{s}^j = (\mathbf{I}^j, \mathbf{Q}^j)$ into each layer of the neural network concurrent with its evaluation, see Fig. 2b and Supplementary Note 5 for details. We have also explored the use of the same type of neural network for quantum state discrimination, in a supervised-learning setting[50–52] (see Supplementary Note 3).

## Training with experimental data

We train the agent based on experimentally acquired episodes to maximize the cumulative reward $R = V_{\text{ver}}/\Delta V - n\lambda$ (see Supplementary Note 4 for details). Here, $V_{\text{ver}}$ is the integrated observation in the final verification measurement $V_{\text{ver}} = \mathbf{w_s s}^{\text{ver}}$ with weights $\mathbf{w_s}$ chosen to maintain the maximal signal-to-noise ratio under Gaussian noise[49,50,53,54]. Therefore, $V_{\text{ver}}/\Delta V$ is a good indicator for the ground-state population, with a normalization factor $\Delta V = \mathbf{w_s}(\langle \mathbf{s}_g \rangle - \langle \mathbf{s}_e \rangle)$ setting the scale. The second term penalizes each cycle by a constant $\lambda$. For larger $\lambda$, trajectories requiring more cycles till termination will

achieve a lower reward. Consequently, the strategy minimizing the averaged reward $\langle R \rangle$ for larger $\lambda$ results in shorter trajectories, i.e. a lower average number of cycles $\langle n \rangle$, while the initialization error $1 - P_g$ is larger. Thus, $\lambda$ controls the trade-off between short episode length and high initialization fidelity. We note that for training and applying the agent, we do not require the explicit functional forms of $\langle n \rangle(\lambda)$ and $(1 - P_g)(\lambda)$, which in general depend on the properties of the quantum system.

We first train the agent to initialize the qubit using fast, high-fidelity readout. In this regime, an initialization strategy based on weighted integration and thresholding is close-to-optimal, and we can thus easily verify and benchmark the strategies discovered by the agent. To study the agent's learning process, we monitor the average number of cycles $\langle n \rangle$ until termination and the initialization error $1 - P_g$, inferred from a fit to the measured distribution of $V_{\text{ver}}$ (see Supplementary Note 2 for details), see Fig. 3a, b. The agent learns how to initialize the qubit for both prepared initial states, starting from either the equilibrium state (red) or its counterpart with populations inverted by a $\pi$-pulse (dark blue). The initialization error $1 - P_g$ converges to about 0.2% after training with only about 30,000 episodes, which includes 100 parameter updates by the reinforcement learning trainer on the PC. The training process takes only three minutes wall clock time. This relatively short training duration, limited mainly by data transfer between the PC and FPGA, enables frequent readjustment of the neural network parameters and thus allows to account for drifts in experimental parameters.

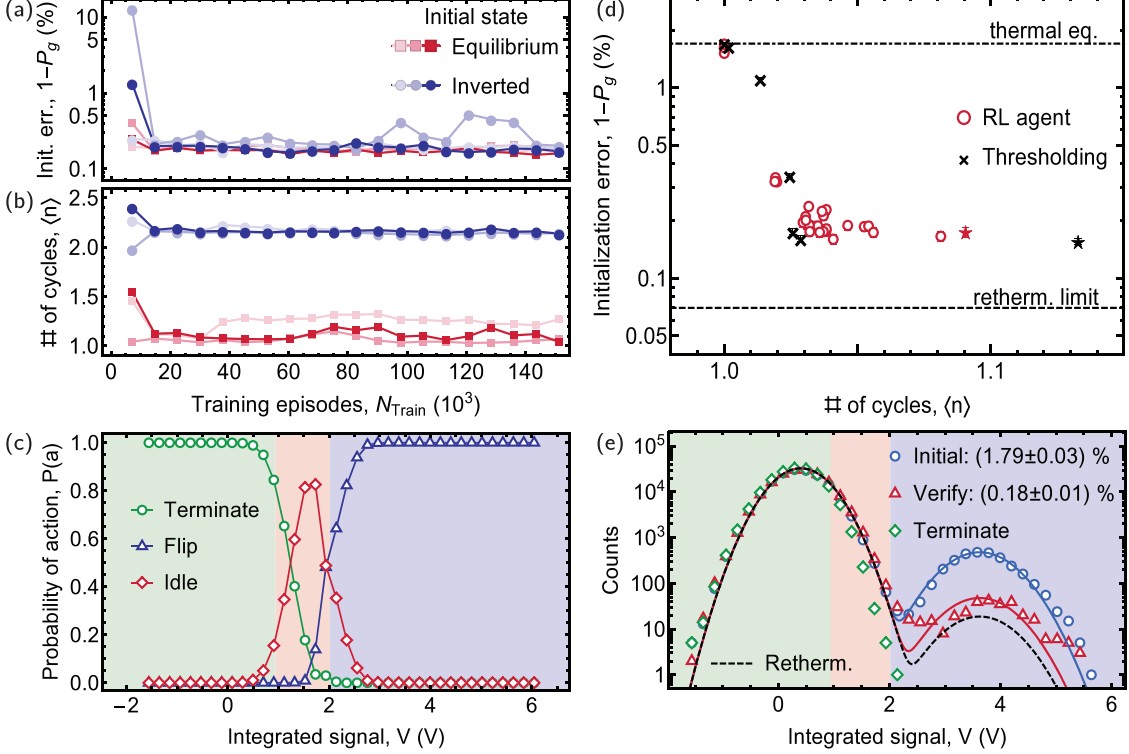

**Fig. 3 | Experimental data for reinforcement learning with a network-based real-time agent. a** Initialization error $1 - P_g$ and **b** average number of cycles $\langle n \rangle$ until termination vs. number of training episodes $N_{\text{Train}}$, when preparing an equilibrium state (red squares) and when inverting the population with a $\pi$-pulse (dark blue circles) for three independent training runs (solid and transparent points). Each datapoint is obtained from an independent validation data set with ~180,000 episodes. **c** Probability of choosing an action $P(a)$ vs. the integrated measurement signal $V$. Actions chosen by the threshold-based strategy are shown as background colors (also for (**e**)). **d** Initialization error $1 - P_g$ vs. average number of cycles $\langle n \rangle$ until termination for an equilibrium state for the reinforcement learning agent (red

circles) and the threshold-based strategy (black crosses). Stars indicate the strategies used for the experiments in (**c**) and (**e**). The dot-dashed black indicates the thermal equilibrium (thermal eq.). Error bars indicate the standard deviation of the fitted initialization error $1 - P_g$. **e** Histogram of the integrated measurement signal $V$ for the initial equilibrium state (blue circles), for the verification measurement (red triangles) and for the measurement in which the agent terminates (green diamonds). Lines are bimodal Gaussian fits, from which we extract ground state populations as shown in the inset. The dashed black line in (**d**) and (**e**) indicates the rethermalization (retherm.) limit (see main text).

## Policy for strong measurements

After the training has been completed, we visualize the agent's strategy by plotting the action probabilities $P(a)$ vs. $V$, see Fig. 3c. We compare this strategy to a thresholding strategy, in which the action is chosen based on the value of $V$ only. We observe that the agent follows this simple strategy in regimes of high certainty. In between, the transitions of the individual probabilities are smooth. This is not due to some deliberate randomization of action choices, but rather a sign that the agent's policy depends on additional information beyond the integrated signal $V$ shown here, as the agent has access to the full measured time trace.

To evaluate the agent's performance we analyze the tradeoff between initialization error $1 − P_g$ and average cycle number $\langle n \rangle$ as a function of the control parameter $\lambda$, see Supplementary Note 4 for details. As expected, we find that an increase in $\langle n \rangle$, controlled by lowering $\lambda$, results in a gain of initialization fidelity until $1 − P_g$ converges to about 0.18% (for $\langle n \rangle \geq 1.1$ cycles), about a tenfold reduction compared to the equilibrium state, see Fig. 3d, e. We attribute the remaining infidelity mostly to rethermalization of the qubit between the termination and the verification cycle, and, possibly, state mixing during the final verification readout. In our experiment, this rethermalization rate is $N_{eq}/T_1 \approx 1$ kHz with $N_{eq} = 1.4\%$, contributing -0.07% to the infidelity. As anticipated, the agent's performance matches the performance of simple, close-to-optimal, thresholding strategies, where we vary the acceptance threshold to control the average cycle number $\langle n \rangle$ (black crosses). This indicates that the strategies discovered by the agent are also close to optimal. In addition, we also note that state transitions are rare, because the measurement time is significantly shorter than the relaxation time $\tau \ll T_1$. Therefore, the ability of the neural network to detect state transitions from the readout time trace does not result in a significant change in performance in the presented experiments. We have also studied the ability of the neural network to distinguish different quantum states in dependence on the measurement time $\tau$ (see Supplementary Note 3) for which we observe pronounced improvements in performance when increasing $\tau$.

## Weak measurements and qutrit readout

The observations until this point demonstrate that our real-time agent performs well and trains reliably on experimentally obtained rewards. Next, we discuss regimes where good initialization strategies are more complex. As a first example, we investigate the agent's strategy and performance when only weakly measuring the qubit. We reduce the power of the readout tone, while keeping its duration and frequency unchanged, such that bimodal Gaussian distributions of a prepared ground and excited state overlap by 25% (see Supplementary Note 2). In this case, we find that the agent benefits from memory, if it is permitted to access information from $l$ previous cycles, see Fig. 4a. Whenever the current measurement hints at the same state as the previous measurement (upper right and lower left in each panel) the agent gains certainty about the state and thus becomes more likely to terminate the process (green region in the lower left corner) or swap the $|g\rangle$ and the $|e\rangle$ state (blue region in the upper right corner). As for strong measurements, we find a trade-off between $\langle n \rangle$ and $1 − P_g$ when varying $\lambda$, see Fig. 4b. Importantly, we observe that agents making use of memory ($l = 2$, red circles) require fewer rounds $\langle n \rangle$ to reach a given initialization error than agents without memory ($l = 0$, green triangles) or a thresholding strategy (black crosses). In addition, we note that the agent without memory ($l = 0$) needs slightly more rounds than the thresholding strategy to reach a certain initialization error, although both methods have an approximately equal amount of information available. We have not investigated this effect in detail, but one possible explanation are decay and rethermalization rates varying during the several days of acquisition time of the data.

In addition, we have studied the performance of the agent when also considering the second excited state $|f\rangle$, which we have neglected

so far. The $|f\rangle$ state is populated with a certain probability due to undesired leakage out of the computational states $|g\rangle$ and $|e\rangle$ during single-qubit, two-qubit and readout operations[54]. Thus, schemes which also reset $|f\rangle$ into $|g\rangle$ are required. For this purpose, we enable the agent to also swap $|f\rangle$ and $|g\rangle$ states by adding a fourth action, and train the agent on a qutrit mixed state with one third $|g\rangle, |e\rangle$ and $|f\rangle$ population. For this qutrit system, state assignment typically processes two different projections of the measurement trace $V = \mathbf{w_V} \mathbf{s}^{ver}$ and $W = \mathbf{w_W} \mathbf{s}^{ver}$, where $\mathbf{w_V}$ and $\mathbf{w_W}$ form an orthonormal set of weights. Here, we use $V$ and $W$ to visualize the agent's strategy. Whenever the measurements firmly indicate that the qutrit is in some given state, the agent proceeds with the corresponding action, while the agent's policy is more complex and harder to predict when measurements fall in-between such clear outcomes, see Fig. 4c.

We find that an agent that can swap $|f\rangle$ to $|g\rangle$, in addition to the other actions, efficiently resets the transmon from a qutrit mixed state with an initialization error $1 − P_g \approx 0.2\%$ for an average number of cycles $\langle n \rangle \approx 2$ (blue squares), see Fig. 4d. In contrast, an agent which cannot access the $gf$-flip action needs significantly more rounds till termination to reach a similar initialization error, as the agent needs to rely on decay from the $|f\rangle$ level, which in our setup had a lifetime of $T_1^{(f)} = 6\,\mu s$. For the agent that cannot access the $gf$-flip action, we also observe a sudden increase in $\langle n \rangle$ from 2.2 to 3.4 when decreasing $\lambda$ from 0.22 to 0.10. Above $\lambda > 0.1$, the agent only resets the $|e\rangle$ level, as the loss in $R$ associated with the additionally required cycles would be larger than the gain associated with the increase in initialization fidelity from resetting the $|f\rangle$ level.

These examples demonstrate the versatility of the reinforcement learning approach to discovering state initialization strategies under a variety of circumstances.

## Discussion

In conclusion, we have implemented a real-time neural-network agent with a sub-microsecond latency enabled by a network design which accepts data concurrently with its evaluation. The need for such optimized real-time control will increase due to the ever more stringent requirements on the fidelities of quantum processes as quantum devices grow in size and complexity. We have successfully trained the agent using reinforcement learning in a quantum experiment and demonstrated its ability to adapt its strategy in different scenarios, including those for which memory is beneficial. Our experiments are an example of reinforcement learning of real-time feedback control on a quantum platform.

While our experiments focused on the initialization of a single qubit into its ground state, it turns out that a range of other conceivable real-time quantum feedback tasks operating on a single qubit are straightforward extensions of the demonstrated protocol. Initialization into an arbitrary superposition state can be achieved by realizing a suitable final unitary operation after qubit initialization. Alternatively, one can perform all measurements in a suitably rotated basis where the target state is one of the measurement basis states. The weak measurement scenario which we explored could be extended as well by measuring in different bases, slowly steering a quantum state towards the desired target without immediate projection.

There are a number of other possible scenarios for real-time quantum feedback control on a single qubit which are less directly related to what we have demonstrated in this work. For example, in the qutrit scenario, one may realize a measurement which does not distinguish between two of the three qutrit states. Realizing such a measurement would enable the detection of decay processes out of that subspace and allow for a subsequent reset into the subspace. One could also extend the presented work to settings in which the qubit is driven, e.g., designing an agent to learn the stabilization of Rabi oscillations, in the spirit of the approach discussed in ref. 55. Finally, in

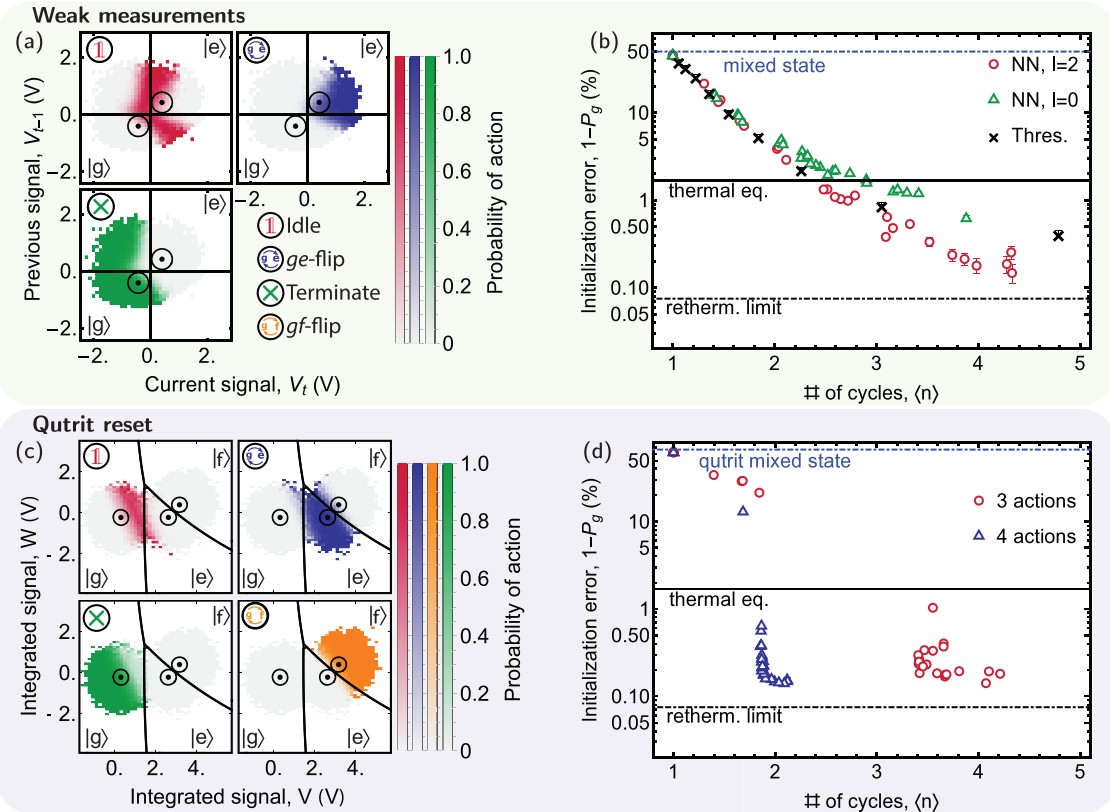

**Fig. 4 | Reinforcement learning results for weak measurements and three-level systems. a** Probability $P(a)$ of selecting the action indicated in the top left corner vs. the signal of the current $V_t$ and the previous $V_{t-1}$ cycle if the agent is permitted to access information from $l = 2$ previous cycles. The radii of the black circles indicate the standard deviation around the means (black dots) of the fitted bi-modal Gaussian distribution. Black lines are the state discrimination thresholds (normalized to 0, see Supplementary Note 2). $P(a)$ is shown for each bin with at least a single count. Empty bins are colored white (also for (**d**)). **b** Initialization error $1 - P_g$ vs. $\langle n \rangle$ for weak measurements for an initially mixed state for $l = 2$ (red circles), $l = 0$ (green triangles) of the neural network (NN) and a thresholding strategy (black crosses). **c** Probability $P(a)$ of choosing the indicated action vs. $V$ and $W$. Black circles indicate the standard deviation ellipse around the means (black dots) of the fitted tri-modal Gaussian distribution. Black lines are state discrimination thresholds (see Supplementary Note 2). **d** Initialization error $1 - P_g$ for a completely mixed qutrit state vs. $\langle n \rangle$ when the agent can select to idle, $ge$-flip and terminate (red circles), and when the agent can in addition perform a $gf$-flip (blue triangles). The dashed black line in (**b**) and (**d**) indicates the rethermalization (retherm.) limit (see main text), the solid black line indicates the thermal equilibrium (thermal eq.). Error bars indicate the standard deviation of the fitted initialization error $1 - P_g$.

the future multi-qubit scenarios can be explored expanding on the techniques presented in this paper.

Understanding the scaling of neural networks with the size of the quantum system and overcoming hardware restrictions on FPGAs are important steps towards applying these methods to larger systems. Such advances will enable the discovery of new strategies for tasks like quantum error correction[36–38] and many-body feedback cooling[31–34].

## Data availability
The data supporting the findings of this letter and corresponding Supplementary Information file have been deposited in the ETH Zurich repository for research data under https://doi.org/10.3929/ethz-b-000637125.

## Code availability
The code used for data analysis is available from the corresponding authors upon request.

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

## Acknowledgements

This work was supported by the Swiss National Science Foundation (SNSF) through the project "Quantum Photonics with Microwaves in Superconducting Circuits" (Grant No. 200021_184686, C.E.), by the European Research Council (ERC) through the project "Superconducting Quantum Networks" (SuperQuNet), by the National Centre of Competence in Research "Quantum Science and Technology" (NCCR QSIT), a research instrument of the Swiss National Science Foundation (SNSF), by ETH Zurich, the Munich Quantum Valley, which is supported by the Bavarian state government with funds from the Hightech Agenda Bayern Plus, and by the Max Planck Society.

## Author contributions

K.R. and J.O. prepared and calibrated the experimental setup. K.R., L.B., and A.A. implemented the neural network on the FPGA. F.M. and C.E. conceived the idea for the experiment. J.L., T.F., and F.M. simulated the system and the network and determined the network structure, training algorithm and reward function, with input from K.R. and C.E. K.R. and J.O. designed the device. G.J.N., M.K., A.R., and J.-C.B. fabricated the device. K.R. and J.O. carried out the experiments and analyzed the data, with support from J.L. K.R., J.L., F.M., and C.E. wrote the manuscript with input from all co-authors. F.M., A.W., and C.E. supervised the work.

## Competing interests

The authors declare no competing interests.
