## [Peer Review File · Nature Communications]

Realizing a Deep Reinforcement Learning Agent for Real-Time Quantum FeedbackREVIEWER COMMENTS

Reviewer #1 (Remarks to the Author):

Kevin Reuer and coauthors report on a neural network approach for real-time feedback control of the status of a superconducting qubit. The reinforcement learning agent is implemented in a field-programmable gate array (FPGA) for real time decision and qubit-control actions, while the parameters of the neural network are processed by a PC and loaded to the FPGA after each cycle of the learning process.

The manuscript is well written, supported by a set of well-defined experiments, and easy to understand for a not specialized reader. The supplementary information helps to gain a deeper understanding of the process and it has enough information to reproduce the findings.

Although the manuscript proves that the model-free reinforcement learning can be used to initialize a qubit in its ground state, it is not clear how this proof-of-concept can be extended to the main claim of “real-time quantum feedback”. Despite the convincing demonstration that the reinforcement learning approach produces similar qubit initialization when compared to the threshold approach, there is no evidence that this approach can be efficiently used on other applications.

My recommendation is to provide more support to the main claim of “real-time quantum feedback” by testing the protocol in a larger set of situations than a qubit (or qutrit) initialization.

As a final comment, I suggest to the authors to comment on the lack of data between 2.2 and 3.4 # of cycles in figure 4(d). Is there a physical or hardware explanation for it?

Reviewer #2 (Remarks to the Author):

The manuscript titled "Realizing a Deep Reinforcement Learning Agent for Real-Time

Quantum Feedback" describes an implementation of a reinforcement learning agent in the FPGA, which is able to make decisions on future actions in less than a microsecond. The main novelty of the manuscript is the introduction and the implementation of a neural network that can consume the qubit state readout traces in almost real-time to produce an action with minimal latency compared to the duration of the readout. The manuscript showcases their implementation with two examples use-cases: qubit state initialization and qubit state discrimination.

I found the manuscript well written and easy to follow. While to my understanding the authors do not show that the reinforcement learning based approach offers direct advantages over traditional approaches in the two example use-cases they present, I still find their results impressive and agree with their main claim: they have demonstrated an efficient approach to use reinforcement learning for making non-trivial decisions almost in real-time, which has applications for example for quantum error-correction or multi-qubit state discrimination. I believe this result will be of interest for the wide audience of Nature comms. and therefore suggest publications once the minor comments below are addressed.

Comments:

* line 61: Add "field-programmable gate array (FPGA)". The acronym is later used without explanation (line 125).

* On line 142 it is mentioned that observations from up to 2 previous cycles can be used. What lead to this choice? Is it due to hard-ware limitations or because no further history is needed?

* Fig 3c seems to indicate that the agent is able to use some of the information in the readout traces to make better actions compared to a reference case of only using fully integrated traces. However, based on Fig. 3d there does not seem to be a significant difference between thresholding and RL agent strategies. Why is that?

* In the fast neural network the later data-points are processed by smaller number of layers. How does this impact the information gained from those data-points? Is this a major limitation of the proposed algorithm? How does the performance of the proposed fast-to-evaluate neural network compare to a one that would have access to all the data-points in

its first layer?

* How λ affects $\langle n \rangle$ is explained quite vaguely. How do the values of $\langle n \rangle$ depend on the value of λ ?

* Fig 4b: why does NN with $l=0$ perform worse than the thresholding strategy? I would have them to be somewhat equivalent or NN to over-perform because the both strategies have in principle equal amount of information available (if I understand correctly, NN has even more because it can access the full readout traces instead of just integrated results).

Reviewer #3 (Remarks to the Author):

This paper implemented a real-time quantum feedback control using deep reinforcement learning with a sub-microsecond latency. This result may be the fastest reinforcement learning agent deployed in a physics experiment so far. Reinforcement learning has demonstrated increasing interest for finding new strategies for many challenging tasks such as quantum error correction, quantum control engineering and many-body feedback cooling. This work successfully trained the agent using reinforcement learning in a quantum experiment and demonstrated its ability to adapt its strategy in different scenarios. This work has potential to generate significant and positive impact on implementation of real-time feedback on quantum platforms and present a timely contribution to this emerging area of machine learning for quantum experiments. Overall, this paper was well written and organized. The following points could be further clarified to improve the readability.

1. The design of reward is critical in reinforcement learning, which directly affects the learning performance. In the Supplementary Material, the authors designed the reward $r_t = (U_{t+1} - U_t)(U_g - U_e) - \lambda$. Firstly, this work also involves the use of unitary gate operation. The notation U is frequently used for unitary operation in quantum information community. Maybe a different notation from U for representing the projected measurement result could be helpful for improving readability. Secondly, in the following explanation of the reward, the authors mentioned “the projected verification measurement U_{n+1} ”, the U_{n+1} seems to be confusing since $t \in \{1, \dots, n\}$. Thirdly, a clearer explanation on what is the measurement result U_t can be useful for experimentalists and readers who want to repeat the training process numerically. Lastly, it will also be helpful to add an

intuitive explanation why such a reward has been chosen and could achieve good performance (this also applies for $R=U_{ver}/\delta U_n/\lambda$) in the main text).

2. The authors employed PPO algorithm. It is worth adding a remark or a sentence why PPO is best suitable for the current task.

3. On page 5 of SI, the authors mentioned “The whole training loop described above is summarized in 1.” There is a typo “in 1”.

For the point-by-point answers, we use the following font conventions:

Italic blue font: comments by the referees

Plain font: answers to the comments of referees

Quotations "...": text from or for the manuscript

Bold font: modifications to the manuscript

Reviewer #1 (Remarks to the Author):

Kevin Reuer and coauthors report on a neural network approach for real-time feedback control of the status of a superconducting qubit. The reinforcement learning agent is implemented in a field-programmable gate array (FPGA) for real time decision and qubit-control actions, while the parameters of the neural network are processed by a PC and loaded to the FPGA after each cycle of the learning process.

The manuscript is well written, supported by a set of well-defined experiments, and easy to understand for a not specialized reader. The supplementary information helps to gain a deeper understanding of the process and it has enough information to reproduce the findings.

We appreciate the reviewer's positive feedback on our work.

Although the manuscript proves that the model-free reinforcement learning can be used to initialize a qubit in its ground state, it is not clear how this proof-of-concept can be extended to the main claim of "real-time quantum feedback". Despite the convincing demonstration that the reinforcement learning approach produces similar qubit initialization when compared to the threshold approach, there is no evidence that this approach can be efficiently used on other applications.

My recommendation is to provide more support to the main claim of "real-time quantum feedback" by testing the protocol in a larger set of situations than a qubit (or qutrit) initialization.

As pointed out correctly by the reviewer, we test the protocol for initialization of a single qubit. We explore initialization in a wide range of different regimes, including strong and weak measurements and qutrit initialization. We understand that the claim "real-time quantum feedback" can be understood more broadly. To clarify that we have only implemented the protocol for a single qubit, we have changed the abstract:

"Here, we have implemented such an agent **for a single qubit** as a sub-microsecond-latency neural network on a field-programmable gate array (FPGA)."

The reviewer is also correctly pointing out that at first sight, qubit/qutrit initialization seems to be limited in scope. However, many quantum feedback applications involving a single qubit can be mapped onto a qubit/qutrit initialization process. An arbitrary superposition state can be prepared by adding one more single qubit pulse after having prepared the ground state or by rotating the measurement basis. Qubit initialization with weak measurements can be extended to steering a quantum state by measuring in a different basis as well.

As many of these additional approaches, while immediately extending the application domain of our experiment, do not require new conceptual steps or techniques, we did not implement them in addition to the demonstrations presented in our manuscript.

In addition, the protocol could be tested on applications involving more than a single qubit. The scaling of the size of the neural network and hardware restrictions are challenges, which we do not

address here. Given the flexibility provided by machine learning techniques, we are confident that these challenges can be overcome for many applications in the future and that our manuscript establishes a first important step in this direction.

We extended our conclusion in the manuscript to better show the connection between our approach and other applications and to make the challenges when scaling to more qubits more apparent (lines 341-375):

“While our experiments focused on the initialization of a single qubit into its ground state, it turns out that a range of other conceivable real-time quantum feedback tasks operating on a single qubit are straightforward extensions of the demonstrated protocol. Initialization into an arbitrary superposition state can be achieved by realizing a suitable final unitary operation after qubit initialization. Alternatively, one can perform all measurements in a suitably rotated basis where the target state is one of the measurement basis states. The weak measurement scenario which we explored could be extended as well by measuring in different bases, slowly steering a quantum state towards the desired target without immediate projection.

There are a number of other possible scenarios for real-time quantum feedback control on a single qubit which are less directly related to what we have demonstrated in this work. For example, in the qutrit scenario, one may realize a measurement which does not distinguish between two of the three qutrit states. Realizing such a measurement would enable the detection of decay processes out of that subspace and allow for a subsequent reset into the subspace. One could also extend the presented work to settings in which the qubit is driven, e.g. designing an agent to learn the stabilization of Rabi oscillations, in the spirit of the approach first discussed in Ref. [53]. Finally, in the future multi-qubit scenarios can be explored expanding on the techniques presented in this paper. Understanding the scaling of neural networks with the size of the quantum system and overcoming hardware restrictions on FPGAs are important steps towards applying these methods to larger systems. Such advances will enable the discovery of new strategies for tasks like quantum error correction [36–38] and many-body feedback cooling [31–34].”

Performing experiments on multiple qubits is unfortunately out of the scope of this manuscript, both given the length of the manuscript and the large experimental effort required.

As a final comment, I suggest to the authors to comment on the lack of data between 2.2 and 3.4 # of cycles in figure 4(d). Is there a physical or hardware explanation for it?

As the reinforcement learning agent selects the termination action based on the observation and its parameters θ , we do not directly control the average number of cycles $\langle n \rangle$. Instead, we modify the parameter λ in the definition of the reward R , thereby controlling the trade-off between initialization error and average length of the initialization sequence. The relationship between λ and $\langle n \rangle$ depends on the parameters of the quantum system.

To make this more apparent, we extended the manuscript (lines 170-181):

“The second term penalizes each cycle by a constant λ . For larger λ , trajectories requiring more cycles till termination will achieve a lower reward. Consequently, the strategy minimizing the averaged reward $\langle R \rangle$ for larger λ results in shorter trajectories, i.e. a lower average number of cycles $\langle n \rangle$, while the initialization error $1 - P_g$ is larger. Thus, λ controls the trade-off between short episode length and high initialization fidelity. We note that for training and applying the agent, we do not require the explicit functional forms of $\langle n \rangle(\lambda)$ and $(1 - P_g)(\lambda)$, which depend on the properties of the quantum system.”

In Fig. 4(d) (also in Fig. 3(d) and 4(b)) λ is swept from 0.001 to 1 with a logarithmic spacing:

$$\lambda = \{0.0010, 0.0022, 0.0046, 0.010, 0.022, 0.046, 0.10, 0.22, 0.46, 1.0\}$$

The agent with three actions (red circles in Fig. 4(d)) initializes the qutrit in about 2.2 cycles for $\lambda = 0.22$, while it takes about 3.4 cycles for $\lambda = 0.10$, indicating a rapid increase in the average number of cycles till termination when increasing λ . While we have not studied this feature in detail, we think the rapid increase is not due to logarithmic discretization of λ , but due to the agent's strategy. The agent with three actions has no option to directly reset the second excited state. For large λ the agent thus only resets the first two levels and does not wait for the f-level to decay, as the punishment λ for executing additional cycles would be too severe. When decreasing the punishment, i.e., λ , below a certain threshold, the agent can suddenly achieve a higher reward by waiting for the second excited state to partially decay, causing a rapid increase in the average number of cycles till termination.

We added a discussion of this aspect to the manuscript (lines 309-321):

"In contrast, an agent which cannot access the gf-flip action needs significantly more rounds till termination to reach a similar initialization error, as the agent needs to rely on decay from the $|f\rangle$ level, which in our setup had a lifetime of $T_1^f = 6 \mu\text{s}$. **For the agent that cannot access the gf-flip action, we also observe a sudden increase in $\langle n \rangle$ from 2.2 to 3.4 when decreasing λ from 0.22 to 0.10. Above $\lambda > 0.1$, the agent only resets the $|e\rangle$ level, as the loss in R associated with the additionally required cycles would be larger than the gain associated with the increase in initialization fidelity from resetting the $|f\rangle$ level."**

Reviewer #2 (Remarks to the Author):

The manuscript titled "Realizing a Deep Reinforcement Learning Agent for Real-Time Quantum Feedback" describes an implementation of a reinforcement learning agent in the FPGA, which is able to make decisions on future actions in less than a microsecond. The main novelty of the manuscript is the introduction and the implementation of a neural network that can consume the qubit state readout traces in almost real-time to produce an action with minimal latency compared to the duration of the readout. The manuscript showcases their implementation with two examples use-cases: qubit state initialization and qutrit state discrimination.

I found the manuscript well written and easy to follow. While to my understanding the authors do not show that the reinforcement learning based approach offers direct advantages over traditional approaches in the two example use-cases they present, I still find their results impressive and agree with their main claim: they have demonstrated an efficient approach to use reinforcement learning for making non-trivial decisions almost in real-time, which has applications for example for quantum error-correction or multi-qubit state discrimination. I believe this result will be of interest for the wide audience of Nature comms. and therefore suggest publications once the minor comments below are addressed.

We appreciate the reviewer's positive feedback on our work and his recommendation for publication.

Comments:

** line 61: Add "field-programmable gate array (FPGA)". The acronym is later used without explanation (line 125).*

We have added the acronym to the manuscript as suggested.

** On line 142 it is mentioned that observations from up to 2 previous cycles can be used. What lead to this choice? Is it due to hard-ware limitations or because no further history is needed?*

For the example cases presented here, we do not expect a significant improvement by considering more cycles. For strong measurements, most of the information is contained in a single measurement and no memory is needed. For weak measurements, the performance difference between considering one cycle and two cycles, was already small. Therefore, we do not expect a significant increase in performance, when considering more cycles.

In our implementation, we are limited by the evaluation time of the pre-processing network on the FPGA. By considering more cycles, the evaluation time of the pre-processing network would increase and would add additional latency. However, we could shorten the evaluation time of the pre-processing network by parallelizing its evaluation.

We added an explanatory note in lines 269-276:

“Importantly, we observe that agents making use of memory ($l = 2$, red circles) require fewer rounds $\langle n \rangle$ to reach a given initialization error than agents without memory ($l = 0$, green triangles) or a thresholding strategy (black crosses). **We have also observed that an agent with $l=1$ only requires about 0.5 more rounds $\langle n \rangle$ than the agent with $l=2$, indicating that there is little additional benefit from considering cycles with $l>2$.**”

We also extended Supplementary Note 5:

“As this information is already available after the previous action was selected, s^j and a^j from l previous rounds are evaluated in a *pre-processing network* (see Fig. 3) while the agent is waiting to receive the most recent readout signal s^t of the current cycle. Thus, no additional latency is introduced to the feedback loop. **However, the evaluation needs to be completed when the most recent readout signal is received. Since evaluating the pre-processing network takes more time when taking more previous rounds l into account, we chose to limit l to two in our implementation. To overcome this limitation in future work, we could parallelize the evaluation of the pre-processing network, enabling it to process information from up to 6 previous cycles.**”

** Fig 3c seems to indicate that the agent is able to use some of the information in the readout traces to make better actions compared to a reference case of only using fully integrated traces. However, based on Fig. 3d there does not seem to be a significant difference between thresholding and RL agent strategies. Why is that?*

The reviewer is correct in pointing out that using some information from the time dependence of the readout traces does not result in a significant difference in performance. In all example cases, the measurement time is short compared to the relaxation time $\tau = 0.256 \mu\text{s} \ll T_1^{(e)} = 13 \mu\text{s}$, such that relaxation/rethermalization events during the measurements are rare. These state transitions cannot be accounted for by weighted integrations. However, all other significant sources of time dependence can be optimally captured by weighted integration, as the underlying correlations in the noise are Gaussian. Therefore, in the limit $\tau \ll T_1^e$, the time dynamics is already mostly captured by weighted integration. The agent can, in addition, detect state transitions, but these events occur too rarely to result in a significant improvement. For longer measurement times τ , the agent might be able to outperform the thresholding approach. While we have not studied the performance of the agent vs measurement time, we studied the performance of a neural network for classification vs measurement time in Supplementary Note 3. Here, we can observe the neural network

outperforming a classifier based on weighted integration and thresholding for longer measurement times due to its ability to detect state transitions.

To better explain this aspect, we have changed the manuscript (lines 234-246):

“This indicates that the strategies discovered by the agent are also close to optimal. **In addition, we note that state transitions are rare, because the measurement time is significantly shorter than the relaxation time $\tau \ll T_1$. Therefore, the ability of the neural network to detect state transitions from the readout time trace does not result in a significant change in performance in the presented experiments. We have also studied the ability of the neural network to distinguish different quantum states in dependence on the measurement time τ (see Supplementary Note 3) for which we observe pronounced improvements in performance when increasing τ .**”

** In the fast neural network the later data-points are processed by smaller number of layers. How does this impact the information gained from those data-points? Is this a major limitation of the proposed algorithm? How does the performance of the proposed fast-to-evaluate neural network compare to a one that would have access to all the data-points in its first layer?*

It is correct that later data points are processed by a smaller number of layers and less information can be extracted from the latest few data points. Before the implementation in the experiment, we have tested the performance of our reinforcement learning approach using simulations. In simulations, we found that a neural network with all input in its first layer did not show a noticeable improvement over the architecture finally used in the experiment.

In addition, evaluating more layers after the last data point was recorded increases the latency of the feedback loop and thus decreases the overall achievable fidelity.

However, we have not tested different neural-network architectures in the experiment.

To make this clearer, we have extended Supplementary Note 5:

“As a result, only the execution of the last layer contributes to the total latency while all other layers are evaluated in parallel with the data acquisition. **We note that data points fed into later layers are processed less than data points fed into earlier layers. In simulations, however, this did not affect the performance of the reinforcement learning agent.**”

** How lambda affects $\langle n \rangle$ is explained quite vaguely. How do the values of $\langle n \rangle$ depend on the value of lambda?*

The parameter λ increases the agent’s reward R for every additional cycle and is used to control the tradeoff between the initialization error $1 - P_g$ and the number of cycles $\langle n \rangle$. The agent is trained to maximize the averaged $\langle R \rangle = \frac{\langle V_{ver} \rangle}{\Delta U} - \lambda \langle n \rangle$ for a given λ . With larger λ longer trajectories achieve a lower averaged reward $\langle R \rangle$ and the optimal solution is to use shorter trajectories (smaller $\langle n \rangle$) at the expense of larger initialization errors. The functional dependence of $\langle n \rangle$ on λ depends on the quantum system under consideration. As we do not require the system model to train and apply our reinforcement learning agent, we did not simulate the system model and thus do not know the exact functional form of $\langle n \rangle(\lambda)$. However, this is no drawback, since in the end we can analyze all results (such as the state initialization fidelity) as a function of $\langle n \rangle$, entirely dropping lambda from the analysis.

We added a paragraph in the manuscript, explaining the relationship between $\langle n \rangle$ and λ (lines 170-181):

“**The second term penalizes each cycle by a constant λ . For larger λ , trajectories requiring more**

cycles till termination will achieve a lower reward. Consequently, the strategy minimizing the averaged reward $\langle R \rangle$ for larger λ results in shorter trajectories, i.e. a lower average number of cycles $\langle n \rangle$, while the initialization error $1 - P_g$ is larger. Thus, λ controls the trade-off between short episode length and high initialization fidelity. We note that for training and applying the agent, we do not require the explicit functional forms of $\langle n \rangle(\lambda)$ and $(1 - P_g)(\lambda)$, which depend on the properties of the quantum system."

** Fig 4b: why does NN with $l=0$ perform worse than the thresholding strategy? I would have them to be somewhat equivalent or NN to over-perform because the both strategies have in principle equal amount of information available (if I understand correctly, NN has even more because it can access the full readout traces instead of just integrated results).*

We did not investigate in detail why the NN with $l=0$ performs slightly worse than the thresholding strategy. The reviewer is right that both have, in principle, the same information available. The NN has also access to the full time trace, although, as outlined in the response to an earlier question, this is not expected to lead to a significant performance improvement in the regime $\tau \ll T_1$.

Possibly, the difference in performance is the result of rethermalization/decay rates varying in time. To accurately resolve the initialization error for weak measurements, a large number of averages was required. It took three days to record the data sets shown in Fig. 4b. The data sets were measured consecutively not interleaved. Rethermalization and decay rates can vary significantly on the timescale of days. However, we would need to investigate these variations in more detail, which was not done for the data set presented in the manuscript.

We added a statement in the manuscript (lines 269-284):

"Importantly, we observe that agents making use of memory ($l = 2$, red circles) require fewer rounds $\langle n \rangle$ to reach a given initialization error than agents without memory ($l = 0$, green triangles) or a thresholding strategy (black crosses). We have also observed that an agent with $l=1$ only requires about 0.5 more rounds $\langle n \rangle$ than the agent with $l=2$, indicating that there is little additional benefit from considering cycles with $l>2$. In addition, we note that the agent without memory ($l=0$) needs slightly more rounds than the thresholding strategy to reach a certain initialization error, although both methods have an approximately equal amount of information available. We have not investigated this effect in detail, but one possible explanation are decay and rethermalization rates varying during the several days of acquisition time of the data."

Reviewer #3 (Remarks to the Author):

This paper implemented a real-time quantum feedback control using deep reinforcement learning with a sub-microsecond latency. This result may be the fastest reinforcement learning agent deployed in a physics experiment so far. Reinforcement learning has demonstrated increasing interest for finding new strategies for many challenging tasks such as quantum error correction, quantum control engineering and many-body feedback cooling. This work successfully trained the agent using reinforcement learning in a quantum experiment and demonstrated its ability to adapt its strategy in different scenarios. This work has potential to generate significant and positive impact on implementation of real-time feedback on quantum platforms and present a timely contribution to this emerging area of machine learning for quantum experiments. Overall, this paper was well written and organized. The following points could be further clarified to improve the readability.

We appreciate the reviewer's positive feedback on our work.

1. The design of reward is critical in reinforcement learning, which directly affects the learning

performance. In the Supplementary Material, the authors designed the reward $r_t = (U_{t+1} - U_t)(U_g - U_e) - \lambda$. Firstly, this work also involves the use of unitary gate operation. The notation U is frequently used for unitary operation in quantum information community. Maybe a different notation from U for representing the projected measurement result could be helpful for improving readability.

We have changed the notation to V , for better readability.

Secondly, in the following explanation of the reward, the authors mentioned “the projected verification measurement U_{n+1} ”, the U_{n+1} seems to be confusing since $t \in \{1, \dots, n\}$.

We thank the reviewer for pointing this out. We changed the $n + 1$ notation in the definition of the reward in Supplementary Note 4 to make this clearer, see in our response to the next point.

Thirdly, a clearer explanation on what is the measurement result U_t can be useful for experimentalists and readers who want to repeat the training process numerically.

The measurement result $V_t = \mathbf{w}_s s^t$ (formerly U_t) is the readout trace s^t the t -th cycle of the measurement integrated with optimal weights \mathbf{w}_s . The weights guarantee a projection onto the axis with maximal signal-to-noise ratio between projected ground and excited states.

While V_t is briefly explained in the main text, a more detailed description of V_t was missing in Supplementary Note 4, on the paragraph defining the reward. We therefore modified Supplementary Note 4 to both define the reward and state the measurement result, now called integrated observation in accordance with the main text, more clearly:

“We choose the reward r^t at time step $t \in \{1, \dots, n-1\}$ as

$$r^t = \frac{V_{t+1} - V_t}{V_g - V_e} - \lambda$$

with the **integrated observation** $V_t = \mathbf{w}_s s^t$ of the t -th **cycle** and a control parameter λ . V_g and V_e are the **average integrated observation** if the qubit is prepared in the ground or excited state. **For the final time step the reward r^n is defined as**

$$r^n = \frac{V_{\text{ver}} - V_n}{V_g - V_e} - \lambda,$$

using the integrated observation in the final verification measurement $V_{\text{ver}} = \mathbf{w}_s s^{\text{ver}}$.”

Lastly, it will also be helpful to add an intuitive explanation why such a reward has been chosen and could achieve good performance (this also applies for $R = U_{\text{ver}} / \Delta U - \lambda$ in the main text).

The chosen reward r^t consists of two terms. The first term $\frac{V_{t+1} - V_t}{V_g - V_e}$ quantifies how much the integrated observation V_{t+1} from the $t + 1$ cycle has moved towards the average ground state observation V_g compared to the previous cycle V_t .

Intuitively, the integrated observation V_t represents the projection of the multidimensional observation s^t , being the measurement time trace, onto the axis intersecting s^g and s^e , the average time traces when preparing the qubit in the ground and excited state, respectively. Assuming white Gaussian noise, projecting onto this axis is optimal, i.e. maximizing the signal-to-noise ratio (see Ref. [48,49,52,53]). Therefore, the term $\frac{V_{t+1} - V_t}{V_g - V_e}$ contains information about the evolution of the qubit state for the network to train on. It is also a simple and easily experimentally obtainable quantity.

The second term λ reduces the reward for additional cycles by a constant amount. Such a term is standard in reinforcement learning. The same reasoning applies for the cumulative reward R .

We have added a sentence in the main text about the choice of the reward R (lines 161-170):
“We train the agent based on experimentally acquired episodes to maximize the cumulative reward $R = V_{\text{ver}}/\Delta V - n\lambda$ (see Supplementary Note 4 for details). Here, V_{ver} is the integrated observation in the final verification measurement $V_{\text{ver}} = \mathbf{w}_s \mathbf{s}^{\text{ver}}$ **with weights \mathbf{w}_s chosen to maintain the maximal signal-to-noise ratio under Gaussian noise [48,49,52,53]. Therefore, $V_{\text{ver}}/\Delta V$ is a good indicator for the ground-state population with a normalization factor $\Delta V = \mathbf{w}_s (\langle \mathbf{s}_g \rangle - \langle \mathbf{s}_e \rangle)$ setting the scale.”**

We also extended the respective paragraph in Supplementary Note 4:
“**The integration weights \mathbf{w}_s are chosen to maximize signal-to-noise ratio under Gaussian noise [5,8-10]. The noise in the readout signal originates from Gaussian noise of vacuum fluctuations, losses in the readout line, and added noise of amplifiers [9]. Therefore, $(V_{t+1} - V_t)/(V_g - V_e)$ is a good indicator for the progress of the initialization compared to the previous round, as and gives direct information if the action at resulted in a quantum state closer to the target state. The parameter λ penalizes every action and thus controls the trade-off between average episode length and initialization fidelity.”**

2. The authors employed PPO algorithm. It is worth adding a remark or a sentence why PPO is best suitable for the current task.

We use the Proximal Policy Optimization algorithm (PPO), as it is currently a state-of-the-art algorithm that is widely used in the machine learning community, is known to give very good results for many different problems without fine-tuning, and yields excellent training results for a variety of architectures and tasks. We have, however, not studied other training algorithms. It is likely that a range of other training algorithms will result in a similar performance.

We added a remark in Supplementary Note 4:
“For the training step we use the Proximal Policy Optimization (PPO) algorithm [13] from the Python library Stable Baselines [14]. **We choose PPO as a state-of-the-art algorithm widely used in the machine learning community.”**

3. On page 5 of SI, the authors mentioned “The whole training loop described above is summarized in 1.” There is a typo “in 1”.

We have fixed the typo on page 5 of the Supplementary Information.

REVIEWERS' COMMENTS

Reviewer #2 (Remarks to the Author):

I thank the authors for carefully addressing all my comments. I recommend publication.

Reviewer #3 (Remarks to the Author):

The quality and readability of this paper have been improved.